# Temperature Dependency Model in Pressure Measurement for the Motion-Capturing Pressure-Sensitive Paint Method

**DOI:** 10.3390/s23249714

**Published:** 2023-12-08

**Authors:** Daiki Kurihara, Hirotaka Sakaue

**Affiliations:** Department of Aerospace and Mechanical Engineering, University of Notre Dame, Notre Dame, IN 46556, USA; dkurihar@nd.edu

**Keywords:** pressure-sensitive paint, temperature dependency, motion-capturing PSP method, analytical model

## Abstract

Pressure-sensitive paint (PSP) has received significant attention for capturing surface pressure in recent years. One major source of uncertainty in PSP measurements, temperature dependency, stems from the fundamental photophysical process that allows PSP to extract pressure information. The motion-capturing PSP method, which involves two luminophores, is introduced as a method to reduce the measurement uncertainty due to temperature dependency. A theoretical model for the pressure uncertainty due to temperature dependency is proposed and demonstrated using a static pressure measurement with an applied temperature gradient. The experimental validation of the proposed model shows that the motion-capturing PSP method reduces the temperature dependency by 37.7% compared to the conventional PSP method. The proposed model also proves that a PSP with zero temperature dependency is theoretically possible.

## 1. Introduction

Surface pressure measurement is a fundamental component in experimental fluid dynamics studies. Traditional measurement techniques using piezoelectric pressure sensors are well established and have been used extensively for decades. However, such sensors acquire pressure information at a single point, which is insufficient for comprehensive study of a flow field. Pressure-sensitive paint (PSP) has received significant attention as a nonintrusive method for surface pressure measurements [1,2,3,4,5,6,7,8,9,10,11,12,13,14]. The luminescent intensity of PSP is sensitive to the partial pressure of oxygen in a test gas due to a photophysical process called oxygen quenching [15]. Due to nonradiative processes during the relaxation of an excited luminescent molecule [16], PSP is also sensitive to temperature, which adds uncertainty to the pressure measurement. This uncertainty is called the temperature dependency. Because the luminescent intensity of PSP changes with temperature as well as with pressure, an image-acquisition device cannot distinguish the change in the intensity as being due to one or the other, which causes uncertainty in the pressure measurement. PSP measurements are conducted for subsonic, supersonic, and/or hypersonic regimes. In the high-speed flow regimes, there are higher pressure changes so that it is more feasible. However, the subsonic regime becomes more challenging since there is a small pressure change, the measurement of which is easily contaminated by the temperature dependency. In the subsonic regime, the pressure changes typically up to 20 kPa, and the temperature variation on the PSP surface is within the ±5 °C range [17]. Even under a several °C temperature change, the temperature uncertainty is significant [17,18]. Liu et al. introduced a model describing factors that influence temperature dependency in the intensity-based PSP method [19]. To reduce the temperature dependency, several approaches have been taken. One of them is a separate surface temperature measurement using IR imaging or a temperature-sensitive paint (TSP) [1,17,20,21,22,23,24]. Assuming that the temperature distribution measured by the separate temperature measurement is identical to the one during the PSP measurement, the temperature dependency of the PSP measurement is compensated based on the temperature distribution. Because of the assumption of the identical temperature distribution, this temperature correction method cannot be used for transient measurements. A second approach uses combined PSP and TSP to correct the temperature dependency [25,26,27]. This measurement requires two image acquisitions for PSP and TSP, which are usually performed by using two cameras. This method can be applied to transient measurements, but the temperature correction procedures can be complex due to the need for additional image processing for the TSP and/or complicated image registration. A third approach involves using specific luminophores in PSP to limit the temperature dependency. Pyrene-based PSP, for example, has been studied as a PSP candidate that does not require an additional temperature measurement. It has a particular wavelength region within its emission spectrum that is pressure sensitive but temperature independent [28,29]. By acquiring the PSP image for this wavelength region, a temperature-independent image from the pyrene-based PSP can be captured. However, due to the limited spectral output range, the signal-to-noise ratio tends to be low. Additionally, it can be difficult to isolate only a particular wavelength of light in various experimental measurements.

Sakaue et al. introduced the motion-capturing PSP method, which allows for moving, fluttering, and free-flight measurements [30,31,32,33]. It involves two luminophores: one that is sensitive to pressure (signal) and another that is insensitive to pressure (reference). The emissions from the signal and reference luminophores are captured by means of a color camera. The motion-capturing PSP method, therefore, only requires a color camera, which reduces the complexity of image processing. The spectra of a two-color PSP can be found in Appendix F. By taking the ratio of the signal and reference images, the pressure distribution can be extracted. If these luminophores have the same behavior with respect to changes in temperature, it is hypothesized that the temperature dependency of the ratio image will be cancelled or reduced (Figure 1).

The reduction in temperature dependency given by the motion-capturing PSP method is investigated herein via both theoretical and experimental approaches. A linear model of the temperature dependency for the motion-capturing PSP method is proposed based on the Taylor series. The linear modeling approach was chosen by considering the following. The temperature range in the subsonic regime is typically small enough to assume linearity. An interpretable model is desired to understand the temperature dependency from the PSP characteristic point of view. The model can be extended if it is applied to a larger temperature range by including higher-order terms, and its accuracy can be improved. However, the model then becomes too complicated to be interpreted. The proposed model is validated via temperature calibration and luminescent imaging. The model relates the pressure uncertainty due to temperature with respect to the pressure sensitivity and the temperature dependency, which are the key factors in pressure and temperature calibrations.

## 2. Background

The pressure and temperature ranges were chosen such that a linear relationship holds in the pressure and temperature calibrations. There are two luminophores involved in the motion-capturing PSP method. Each luminophore emits light with a given luminescent intensity. This luminescent intensity is captured using the intensity method [34]. The relationship between the intensity, I, and pressure, P, is given as
(1)$$\frac{I_{q,ref}}{I_{q}}=A_{q}+B_{q}\frac{P}{P_{ref}},\;\;q=\left(R,S\right)$$

The subscript q can be either S or R, indicating the signal or reference luminophore, respectively. The subscript ref denotes a quantity under the reference conditions. A and B are calibration coefficients determined by the pressure calibration. The calibration coefficients based on the intensity method of signal and reference luminophores have the relationship shown in Equation (2) because the Stern–Volmer equation is equal to unity under the reference conditions.
(2)$$A_{q}+B_{q}=1\;(0\leq A_{q},\;B_{q}\leq 1)$$

The intensity ratio is described as a linear function of the temperature, T, for a given temperature range:(3)$$\frac{I_{q}}{I_{q,ref}}={c_{0}}_{q}+{c_{1}}_{q}\frac{T}{T_{ref}}$$
where $c_{0}$ and $c_{1}$ are calibration coefficients determined by the temperature calibration. Equation (3) is valid for the considered temperature range in this paper. Under the reference conditions, the above equations are equal to unity, and so the same restrictions for the calibration coefficients as in Equation (2) apply.
(4)$${c_{0}}_{q}+{c_{1}}_{q}=1$$

For the motion-capturing PSP method, the signal and reference intensities are ratioed [32]. The following relationship between the intensity ratio, $I_{R}/I_{S}$, and the pressure ratio, $P/P_{ref}$, is used in pressure measurements:(5)$$\frac{1}{\alpha }\frac{I_{R}}{I_{S}}=A_{M}+B_{M}\frac{P}{P_{ref}}$$
where α is a constant based on the intensity ratio under the reference conditions: $\alpha =I_{R,ref}/I_{S,ref}$. The subscript M denotes the motion-capturing PSP method. Equation (5) is equal to unity under the reference conditions.
(6)$$A_{M}+B_{M}=1\;\left(0\leq A_{M},\;B_{M}\leq 1\right)$$

In temperature measurements, the following relationship is used to determine the temperature ratio, $T/T_{ref}$, from the intensity ratio, $I_{S}/I_{R}$:(7)$$\alpha \frac{I_{S}}{I_{R}}={c_{0}}_{M}+{c_{1}}_{M}\frac{T}{T_{ref}}$$

Equation (7) is equal to unity under the reference conditions.
(8)$${c_{0}}_{M}+{c_{1}}_{M}=1$$

As Equation (3) is valid for the considered temperature range, Equation (7) is also valid for the range.

## 3. Temperature Dependency Model in Pressure Measurement

In pressure measurement using a PSP, the pressure sensitivity, σ, converts the luminescent intensity of a PSP to a pressure value. The value of σ can be defined as the slope of the pressure calibration under the reference conditions. Similarly, the temperature dependency of a PSP, δ, can be defined as the slope of the temperature calibration under the reference conditions. By taking the derivatives of Equations (1), (3), (5) and (7), σ and δ for the motion-capturing PSP method and the intensity method of signal and reference luminophores are given as follows:(9)$$\sigma_{q}=\frac{𝜕}{𝜕P}\left(\frac{I_{q,ref}}{I_{q}}\right)=\frac{B_{q}}{P_{ref}}$$
(10)$$\sigma_{M}=\frac{𝜕}{𝜕P}\left(\frac{1}{\alpha }\frac{I_{S}}{I_{R}}\;\right)=\frac{B_{M}}{P_{ref}}$$
(11)$$\delta_{q}=\frac{𝜕}{𝜕T}\left(\frac{I_{q}}{I_{q,ref}}\;\right)=\frac{{c_{1}}_{q}}{T_{ref}}$$
(12)$$\delta_{M}=\frac{𝜕}{𝜕T}\left(\alpha \frac{I_{S}}{I_{R}}\right)=\frac{{c_{1}}_{M}}{T_{ref}}$$

In general, $\delta_{q}$ is a negative value due to the nonradiation process of a luminophore [32].

The pressure calibration for the motion-capturing PSP method given in Equation (5) can also be described as a ratio of Equation (1) for the signal and reference luminophores:(13)$$\frac{1}{\alpha }\frac{I_{R}}{I_{S}}=\frac{I_{R}}{I_{R,ref}}\frac{I_{S,ref}}{I_{S}}=\frac{A_{S}+B_{S}\cdot P/P_{ref}}{A_{R}+B_{R}\cdot P/P_{ref}}$$

Using a Taylor series expansion under the reference conditions, Equation (13) can be described as a linear relationship with higher-order terms. When $B_{R}=0$, Equation (13) becomes linear. Assuming that $B_{R}\ll 1$, by neglecting the higher-order terms and using unity equations, the following form can be obtained:(14)$$\begin{array}{cc} \frac{1}{\alpha }\frac{I_{R}}{I_{S}} & =A_{S}-A_{R}+1+\left(B_{S}-B_{R}\right)\frac{P}{P_{ref}}\\ & =A_{M}+B_{M}\frac{P}{P_{ref}} \end{array}$$

A detailed derivation can be found in Appendix A. Equation (14) describes the relationship between the coefficients for the motion-capturing PSP method and the intensity method. If pressure-independent $I_{R}$ is assumed, an equation identical to that discussed in [31] is obtained.
(15)$$B_{M}=B_{S}-B_{R}$$

From Equation (15), it can be seen that $\sigma_{M}$ is the difference between $\sigma_{S}$ and $\sigma_{R}$.
(16)$$\sigma_{M}=\sigma_{S}-\sigma_{R}$$

Ideally, the intensity method for the reference luminophore is independent of pressure. In this case, $\sigma_{R}=0$ and so $\sigma_{M}=\sigma_{S}$.

For the motion-capturing PSP method, the temperature calibration can also be described as a ratio of Equation (3) for the signal and reference luminophores:(17)$$\frac{I_{S}}{I_{S,ref}}\frac{I_{R,ref}}{I_{R}}=\alpha \frac{I_{S}}{I_{R}}=\frac{{c_{0}}_{S}+{c_{1}}_{S}\cdot T/T_{ref}}{{c_{0}}_{R}+{c_{1}}_{R}\cdot T/T_{ref}}$$

Similarly, Taylor series expansion under the reference conditions is used to linearize Equation (17). By neglecting the higher-order terms and using the unity equations, the following relationships can be found.
(18)$$\begin{array}{cc} \alpha \frac{I_{S}}{I_{R}} & ={c_{0}}_{S}-{c_{0}}_{R}+1+\left({c_{1}}_{S}-{c_{1}}_{R}\right)\frac{T}{T_{ref}}\\ & ={c_{0}}_{M}+{c_{1}}_{M}\frac{T}{T_{ref}} \end{array}$$
(19)$${c_{1}}_{M}={c_{1}}_{S}-{c_{1}}_{R}$$
(20)$$\delta_{M}=\delta_{S}-\delta_{R}$$

A detailed derivation can be found in Appendix B. Equation (20) indicates that $\delta_{M}$ is the difference between $\delta_{S}$ and $\delta_{R}$. If $\delta_{S}$ and $\delta_{R}$ are the same, the motion-capturing PSP method has zero temperature dependency.

By solving Equation (5) for the measured pressure, P, it can be described in terms of the reference pressure, calibration coefficients, and intensities.
(21)$$P=\frac{P_{ref}}{B_{M}}\left(\frac{1}{\alpha }\frac{I_{R}}{I_{S}}-A_{M}\right)$$

Taylor series expansion under the reference conditions is used to linearize Equation (21). Since a PSP is both pressure and temperature sensitive, it is necessary to take partial derivatives with respect to each quantity:(22)$$P=P_{ref}+\frac{𝜕}{𝜕T}{\left.\left[\frac{P_{ref}}{B_{M}}\left(\frac{1}{\alpha }\frac{I_{R}}{I_{S}}-A_{M}\right)\right]\right|}_{ref}∆T+\frac{𝜕}{𝜕P}{\left.\left[\frac{P_{ref}}{B_{M}}\left(\frac{1}{\alpha }\frac{I_{R}}{I_{S}}-A_{M}\right)\right]\right|}_{ref}\Delta P+H.O.T.$$
where ∆T is temperature change and ∆P is pressure change. By neglecting the higher-order terms and evaluating derivatives under the reference conditions, the expression for the measured pressure in terms of ∆T and ∆P is derived as follows. Detailed derivations are given in Appendix C.
(23)$$P=P_{ref}-\frac{\delta_{M}}{\sigma_{M}}∆T+∆P$$

Here, the temperature uncertainty factor is defined as the ratio of the temperature dependency and the pressure sensitivity:(24)$$\zeta_{M}=\frac{\delta_{M}}{\sigma_{M}}$$

The pressure uncertainty due to temperature, $P_{u,M}$, is now defined as
(25)$$P_{u,M}=-\zeta_{M}∆T$$
(26)$$P=P_{ref}+P_{u,M}+∆P$$

Equation (23) states that the measured pressure, P, is the sum of the reference pressure and the actual pressure change if there is no temperature change during a PSP measurement. If ∆T is nonzero during the PSP measurement, the second term in Equation (23) needs to be considered, which is the pressure uncertainty due to the temperature. Equations (25) and (26) indicate that a high pressure sensitivity and a low temperature dependency yield a low temperature uncertainty factor, which results in a lower pressure uncertainty. As described in Equation (20), the temperature dependency of the motion-capturing PSP method is zero when the temperature dependencies of the intensity method for the signal and reference luminophores are equal.

For the intensity method, only a signal luminophore is used. It gives the intensity method of the reference luminophore as ${c_{1}}_{R}=0$. This reduces Equations (24) to (26) as follows.
(27)$$\zeta_{S}=\frac{\delta_{S}}{\sigma_{S}}$$
(28)$$P=P_{ref}+P_{u,M}+∆P_{u,S}=-\zeta_{S}∆T$$
(29)$$P=P_{ref}+P_{u,S}+∆P$$

The pressure uncertainty due to temperature is given in the same form as in Equation (25). The difference between the motion-capturing PSP and the intensity methods comes from the value of the pressure sensitivity and the temperature dependency to give the temperature uncertainty factor. The proposed model applies to other formulations of two-color PSPs as long as the nonlinearity of the two luminophores with respect to pressure and temperature is negligible.

### Model Analysis

The value of σ is between 0 and 1, whereas δ can be a positive or negative value [28]. Based on these restrictions on σ and δ, five different cases can be considered (Table 1). For an ideal case (*case 1*), where the signal and reference luminophores have the same temperature dependency, ${\delta_{M}=\delta }_{S}-\delta_{R}=0$. In this case, the temperature dependency of the motion-capturing PSP method, $\delta_{M}$, is zero. The pressure uncertainty due to temperature, $\zeta_{M}$, is also zero. In general, the temperature dependencies of luminophores are negative (*case 2*) [34]. With the condition $\left|\delta_{R}\right|<2\left|\delta_{S}\right|$, this case results in a reduction in the temperature dependency for the motion-capturing PSP method. *Cases 3*, *4*, and *5* are nonstandard for experimental measurements because most PSP luminophores exhibit negative temperature dependencies. However, if the temperature dependency of each luminophore is opposite in sign, as for *cases 3* and *4*, this results in an increase in temperature dependency. The temperature dependency of the motion-capturing PSP method, however, can be reduced if the intensity method for both luminophores has a positive temperature dependence (*case 5*). As for *case 2*, *case 5* also needs to satisfy the condition $\left|\delta_{R}\right|<2\left|\delta_{S}\right|$ in order to reduce the temperature dependency.

As an illustrative description, Figure 2 shows the pressure uncertainty due to temperature for the five different cases based on Equations (26) and (29). Pressure sensitivities and temperature dependencies were chosen based on the experimental validation discussed in Section 4. The pressure sensitivities of the signal and reference luminophores obtained via the intensity method were $\sigma_{S}=0.570$ and $\sigma_{R}=0.083$%/kPa for all cases. The temperature dependencies of the signal and reference luminophores obtained via the intensity method were $\delta_{S}=-0.730$ and $\delta_{R}=-0.340$, also shown in Figure 2. As discussed, *cases 3*, *4*, and *5* are impractical in general. For *case 2*, the motion-capturing PSP method shows lower pressure uncertainty due to temperature compared to that from the intensity method. For the ideal *case 1*, the pressure uncertainty due to temperature for the motion-capturing PSP method is zero.

## 4. Experimental Validation

For a two-color PSP, two luminophores were chosen: tris(4,7-diphenyl-1,10-phenanthroline) ruthenium (II) dichloride complex (Alfa Aesar) as a pressure-sensitive luminophore and fluorescein (Tokyo Chemical Industry) as a pressure-insensitive luminophore. A luminescent solution was prepared by dissolving the pressure-sensitive and -insensitive luminophores in dichloromethane with concentrations of 0.3 mM and 0.5 mM, respectively. Polymer (RTV rubber, KE-42, Shin-Etsu Silicone) and ceramic (Silica gel, Sigma-Aldrich) were then added into the luminescent solution. The polymer content was 40 w%. An ultrasonic bath was used to dissolve the polymer for 10 min. The two-color PC-PSP was then coated over an aluminum block with seven thermocouples for temperature monitoring. An a priori pressure calibration was performed to determine the pressure sensitivities $\sigma_{M}$, $\sigma_{S}$, and $\sigma_{R}$, which were 0.493 ± 0.007, 0.570 ± 0.005, and 0.083 ± 0.004%/kPa, respectively. The method for obtaining the pressure sensitives is described in Appendix D. Error propagation of the fitting uncertainty and repeatability was used to estimate the pressure calibration uncertainty. Figure 3 shows a schematic of the validation setup. An aluminum block coated with the two-color PSP was placed between a Peltier heater and a cooler. A purple LED (Thorlabs inc. SOLIS-405C) excited the two-color PSP at 405 nm. A scientific color camera (NAC image technology HX-7s) captured luminescent intensities from the two-color PSP as red and green images. The signal intensity was acquired as a red image, and the reference intensity was acquired as a green image. The camera’s spectral sensitivities have an overlap region between the green and red channels. Light in this region is accounted as both channels, which is called “crosstalk”. This effect was calibrated by the pressure and temperature calibrations. An optical high-pass filter (470 nm) was placed in front of the camera to eliminate the excitation light. By using the Peltier heater and the cooler, a linear temperature distribution of approximately ±4 ℃ in range was created with a reference temperature at 23.66 ± 0.02 ℃. The temperature at the thermocouple locations was monitored using a data logger (HIOKI LR8431-20). The temperature profiles obtained during the experimental validation can be found in Appendix E.

Since the atmospheric pressure was constant during the validation test, the intensity change was only dependent on the temperature. Using the intensity ratio at the thermocouple locations, the in situ temperature calibration was obtained (Figure 4). The values of $\delta_{M}$, $\delta_{S}$, and $\delta_{R}$ obtained during the validation test were −0.397 ± 0.012, −0.731 ± 0.004, and −0.340 ± 0.004%/°C, respectively. The error propagation of the fitting uncertainty was used to estimate the temperature calibration uncertainty.

Figure 5 shows the pressure uncertainty over the PSP surface given by (a) the motion-capturing PSP method and (b) the intensity method. The temperature profile along the line A–B is shown in Figure 5c and is linear, as expected from the experimental setup. Pressure uncertainties also exhibited a linear gradient in the horizontal direction and a constant pressure uncertainty distribution in the vertical direction. It can be seen that the magnitude of the pressure uncertainty was reduced by applying the motion-capturing PSP method. To quantify the reduction in the uncertainty, the pressure uncertainty model given in Equations (26) and (29) was compared to the measured pressure uncertainty profiles (Figure 6). The pressure uncertainty model and its uncertainty are shown as dashed lines and colored areas, respectively. The uncertainties of the model were estimated from the temperature measurement and PSP calibration errors. The measured pressure uncertainty agreed with the proposed model and was within the model’s uncertainty.

Table 2 summarizes the calibration coefficients. The pressure sensitivity and temperature dependency of the motion-capturing PSP method based on Equations (16) and (20) were 0.487 ± 0.007%/kPa and −0.391 ± 0.017%/°C, which agreed with the calibration results. Based on these coefficients, uncertainty factors were listed using Equations (24) and (27). The intensity method for the signal luminophore resulted in a higher pressure sensitivity and temperature dependency, as compared with those obtained via the motion-capturing PSP method. The motion-capturing PSP method showed lower pressure sensitivity and temperature dependency. The motion-capturing PSP method showed, however, a lower uncertainty factor than did the intensity method for the signal luminophore. The motion-capturing PSP method reduced the uncertainty factor by 37.7 ± 2.7%, as compared to the intensity method.

## 5. Conclusions

Pressure-sensitive paint (PSP) has been utilized to capture surface pressure in recent years. Due to the nature of the photophysical processes inherent to the functionality of PSP, a temperature dependency exists that introduces a major measurement uncertainty for pressure measurements. To model the pressure uncertainty due to temperature, the intensity method and motion-capturing PSP method were investigated using theoretical and experimental approaches. The proposed model relates the pressure uncertainty due to temperature with respect to pressure sensitivity and temperature dependency. The model was validated based on experimental results from pressure and temperature calibrations and from the validation experiment. The model showed that 1. the temperature dependency of the motion-capturing PSP method is the difference between the temperature dependencies of the signal and reference luminophores; 2. the motion-capturing PSP method can reduce pressure uncertainty due to temperature, as compared to the intensity method; and 3. the measurement error due to the temperature dependency is described by $\sigma_{M}$, $\delta_{M}$, and ΔT. The model can also be used to select the two luminophores for the motion-capturing PSP method to achieve zero temperature dependency. From the validation experiment, the temperature uncertainty factor of the motion-capturing PSP method was reduced by 37.7 ± 2.7%, as compared to that of the intensity method.

## Figures and Tables

**Figure 1 sensors-23-09714-f001:**
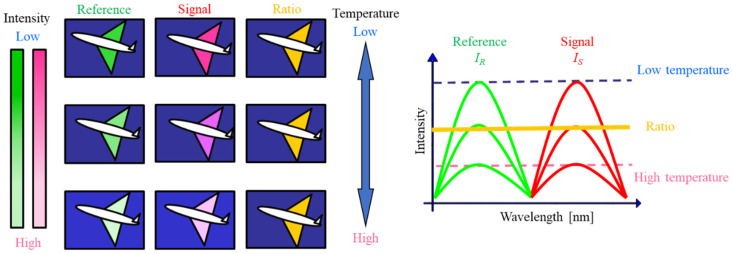
Conceptual description of the temperature dependency for the motion-capturing PSP method.

**Figure 2 sensors-23-09714-f002:**
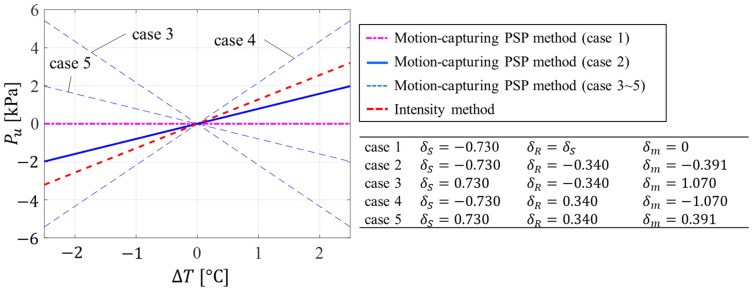
Illustrative example of pressure uncertainty due to temperature for considered cases.

**Figure 3 sensors-23-09714-f003:**
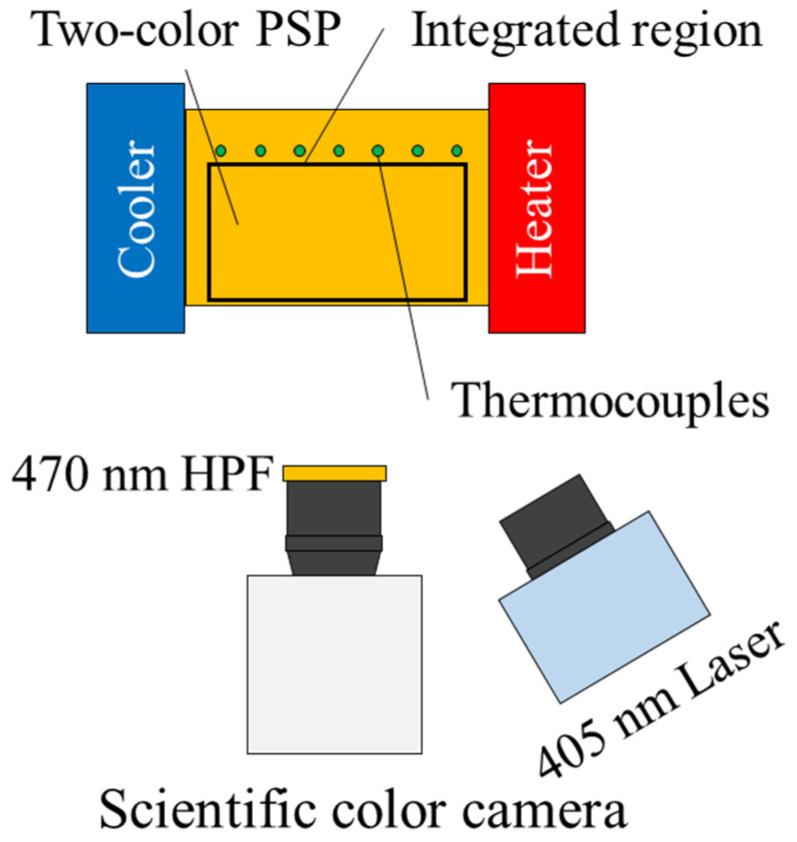
Schematic of the experimental validation setup.

**Figure 4 sensors-23-09714-f004:**
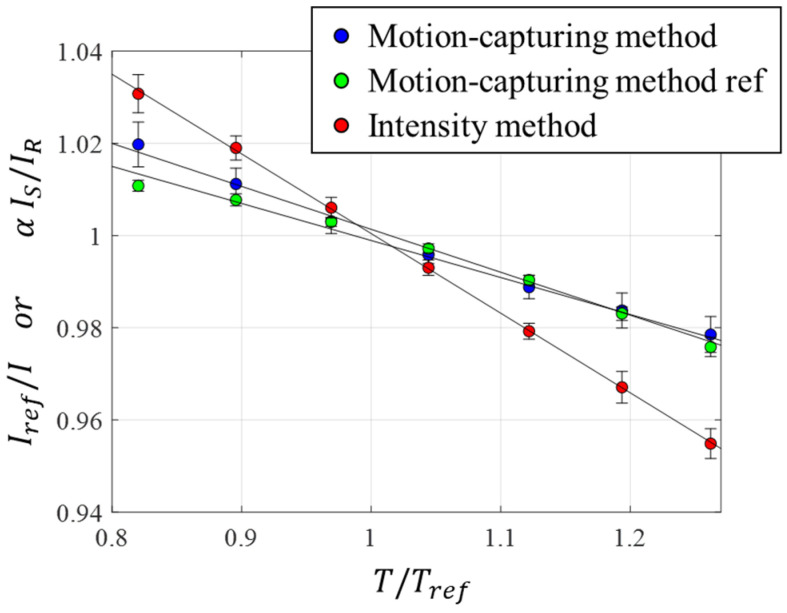
In situ temperature calibration.

**Figure 5 sensors-23-09714-f005:**
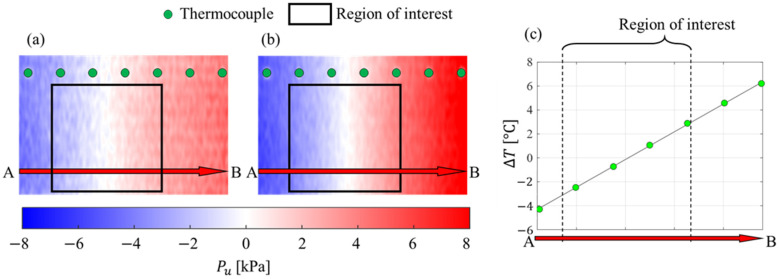
Pressure uncertainty due to temperature distribution based on (**a**) the motion-capturing PSP method and (**b**) the intensity method, and (**c**) the temperature profile along the line A–B.

**Figure 6 sensors-23-09714-f006:**
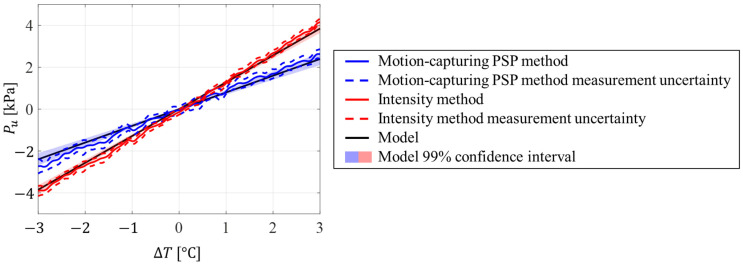
Pressure uncertainty profiles due to temperature based on the motion-capturing PSP method in blue and the intensity method in red. Dashed lines show the model prediction. The blue and red areas are model uncertainties.

**Table 1 sensors-23-09714-t001:** Five different cases of the uncertainty factor in Equations (24) and (27).

*case 1*	$\delta_{R}=\delta_{S}$	$\zeta_{S}>\zeta_{M}=0$	$P_{u,S}<P_{u,M}=0$
*case 2*	$\delta_{R}\leq 0$, $\delta_{S}<0$	$\zeta_{S}\geq \zeta_{M}$	$P_{u,S}\geq P_{u,M}$
*case 3*	$\delta_{R}\leq 0$, $\delta_{S}>0$	$\zeta_{S}\leq \zeta_{M}$	$P_{u,S}\leq P_{u,M}$
*case 4*	$\delta_{R}\geq 0$, $\delta_{S}<0$	$\zeta_{S}\leq \zeta_{M}$	$P_{u,S}\leq P_{u,M}$
*case 5*	$\delta_{R}\geq 0$, $\delta_{S}>0$	$\zeta_{S}\geq \zeta_{M}$	$P_{u,S}\geq P_{u,M}$

**Table 2 sensors-23-09714-t002:** Pressure and temperature calibration coefficients.

Pressure Sensitivity [%/kPa]		Temperature Dependency [%/°C]		Uncertainty Factor [kPa/°C]	
$\sigma_{M}$	0.493 ± 0.007	$\delta_{M}$	−0.397 ± 0.012	$\zeta_{M}$	−0.799 ± 0.031
$\sigma_{S}$	0.570 ± 0.005	$\delta_{S}$	−0.731 ± 0.004	$\zeta_{S}$	−1.282 ± 0.023
$\sigma_{R}$	0.083 ± 0.004	$\delta_{R}$	−0.340 ± 0.004		
$\sigma_{S}-\sigma_{R}$	0.487 ± 0.007	$\delta_{S}-\delta_{R}$	−0.391 ± 0.012		

## Data Availability

Data available on request.

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
