# Peer review of "Temperature Dependency Model in Pressure Measurement for the Motion-Capturing Pressure-Sensitive Paint Method"

_sensors, 2023, doi:10.3390/s23249714_

Round 1

Reviewer 1 Report

Comments and Suggestions for Authors

The manuscript “Temperature Dependency Model in Pressure Measurement for the Motion-Capturing Pressure-Sensitive Paint Method” proposed a temperature dependency analysis for pressure-sensitive paint. The proposed model will be useful for the selection of luminophore for the motion-capturing PSP. Although this is an important work of currently investigated topic, the reviewer recommends “major revision” due to insufficient contents. The authors should check their manuscript before submission.

1.         line 34: “the luminescent intensity of PSP is changes with” should be “the luminescent intensity of PSP changes with”

2.         line 37: “subsonic, supersonic, hypersonic regimes” should be “subsonic, supersonic, and/or hypersonic regimes.”

3.         line 42: “°?” should not be italic font.

4.         line 53: The authors stated that “This measurement requires two image acquisitions for PSP and TSP which are usually performed by using two cameras.” The reviewer disagrees the statement because there are some reports measuring pressure and temperature distribution using a single camera. Then, this is not a drawback of the approach. The approach is also adopted in other two-color measurement techniques such as laser induced fluorescence.

5.         line 64: The reviewer understands that the one of the advantages of the motion-capturing PSP is a measurement by a single color camera. The explanation is insufficient; then, the reviewer recommends adding the explanation.

6.         Equation (1): The variables I and P are not defined.

7.         Equation (3): T is not defined.

8.         line 115: equation (8) may be equation (7).

9.         Equation (9): “I” is a variable depending on both pressure and temperature. Isn't it a partial derivative?

10.     Equation (10) and (11) are identical.

11.     line 129: Please specify the conditions under which a linear approximation is possible. P << P_ref ?

12.     Equation (14) and Appendix A: The same discussion was conducted in Kurihara et al., Aerosp. Sci. Technol., 103, 105878 (2020). Please cite this paper appropriately to avoid self-plagiarism.

13.     Equation (16): sigma_m should by sigma_M.

14.     line 233: The uncertainties are considered to be very small in spite of large error bars in figure 4. It is strange that the uncertainties of the temperature sensitivity are comparable to the uncertainties of the pressure sensitivity, which have much smaller error bars. Please check the calculation. The authors should explain the uncertainty calculations with specific figures.

15.     Conclusions: “Conclusions” is almost identical to the abstract. Please revise it.

Author Response

Please see the uploaded PDF document.

Reviewer 2 Report

Comments and Suggestions for Authors

This paper deals with the temperature sensitivity of pressure-sensitive paints (PSPs), called motion-capture PSPs, which are composed of two different luminescent dyes. One dye is pressure-sensitive, while the other is not. The authors derived the sensitivity from the standard linear pressure or temperature dependence on the emission intensity of the paint.  Owing to the similar temperature dependence of the two dyes, the temperature sensitivity of the ratio of the intensities of the two dyes was substantially lower than the temperature sensitivity of the PSP dye alone.

This paper may be accepted for publication in this journal after appropriate revisions. The authors are expected to consider the following comments when making revisions. 

  1. Please make a symbol table and include it in your manuscript, as there are many symbols used in this paper.

  2. p. 1 line 42 ⁰C Do not italicize C, since it is a temperature unit. The same applies to all the following sentences.

  3. p. 2 Fig. 1, (I think they can be put in other figures, but) please show the spectral measurements of the two dyes actually used in this study.

  4. p. 3, Eq. (8) The subsubscript q may be M. 

  5. p. 4 Eq. (10) This equation should be for temperature sensitivity δ_q. 

  6. p. 4 Eq. (16) The subscript m may be M. 

  7. p. 6, line 192  Add symbols corresponding to the values (0.570, 0.083 %/kPa). Since temperature sensitivity is discussed in the next sentence, please also indicate the value of temperature sensitivity in the text.

  8. p. 7, line 204 Same as comment (2). A diagram of the actual emission wavelengths is welcome to be shown here.

  9. p. 7, line 221 Indicate the sensitivity of the red and green sensors of the camera used and mention the possibility of mixed detection of the emission of the two dyes.

Author Response

Please see the uploaded PDF document.

Reviewer 3 Report

Comments and Suggestions for Authors

The article presents a method to reduce the temperature dependency of pressure-sensitive paint (PSP) measurements using two luminophores with different pressure sensitivities which is called the motion-capturing PSP method. It is based on taking the ratio of the signal and reference intensities from the two luminophores. The result shows that the motion-capturing PSP method can reduce the temperature dependency by 37.7% compared to the conventional intensity method. The article is well-written, clear, and concise.

Some minor suggestions for improvement:

1. In Page4 Equation (10) is exactly the same as Equation (11), one of them may have written incorrectly.

2. It is recommended to supplement the types, characteristics and assembly structures of the two luminescent materials in the two-color PSD structure. It would be useful to explain how the two luminophores were selected and characterized in terms of their pressure sensitivities and temperature dependencies. This would help readers assess the validity and generality of the proposed method.

3. It would be better to discuss the limitations and applicability of the proposed model for different PSP formulations, flow regimes, and temperature ranges.

Author Response

Please see the uploaded PDF document.

Round 2

Reviewer 1 Report

Comments and Suggestions for Authors

The authors have taken into consideration the reviewer's remarks.